# Photo-triggered full-color circularly polarized luminescence based on photonic capsules for multilevel information encryption

Siyang Lin [1], Yuqi Tang [2], Wenxin Kang [1], Hari Krishna Bisoyi [3], Jinbao Guo [1] ✉ & Quan Li [2,3] ✉

Materials with phototunable full-color circularly polarized luminescence (CPL) have a large storage density, high-security level, and enormous prospects in the field of information encryption and decryption. In this work, device-friendly solid films with color tunability are prepared by constructing Förster resonance energy transfer (FRET) platforms with chiral donors and achiral molecular switches in liquid crystal photonic capsules (LCPCs). These LCPCs exhibit photoswitchable CPL from initial blue emission to RGB trichromatic signals under UV irradiation due to the synergistic effect of energy and chirality transfer and show strong time dependence because of the different FRET efficiencies at each time node. Based on these phototunable CPL and time response characteristics, the concept of multilevel data encryption by using LCPC films is demonstrated.

The revolution and development of information technology have significantly changed our lives, and the world has entered the digital era. People are surrounded by massive amounts of information but are barely able to distinguish true information from false information. In the context of this information explosion, information security has attracted considerable attention. The quick response (QR) code is the simplest encryption technology for transforming information into digital data. Advanced materials are the foundation for developing high-tech information encryption, and much effort has been devoted to exploring a variety of information encryption materials[1–5] Currently, the use of fluorescence-switchable materials[6–9] or time-responsive afterglow materials[10–13] is a popular strategy for upgrading QR codes to three-dimensional (3D) codes. Fluorescence is an important factor for information encryption and can act as a metaphorical skin for information. Circularly polarized luminescence (CPL) can afford one more dimension of information (i.e., the state of polarization) than common fluorescence and is of great potential in the field of information encryption[14–17]. For instance, a double-layer pattern was used to reveal hidden CPL information under continuous UV irradiation[18], or the

authenticity of information was verified by using the CPL spectrum[19]. CPL comprises differential intensities of right- and left-handed fluorescence, which directly represents the structural information of chiral systems in the excited state[20–22]. Among the diverse CPL-active materials, luminescent cholesteric liquid crystals (CLCs) have been considered promising candidates for generating tunable CPL with large luminescence dissymmetry factors ($g_{lum}$)[23–27] because of the self-organized helical superstructure[28–34] and stimuli-responsive nature of CLCs[35–42]. Recently, a wavelength-controllable CPL-active laser array was constructed by thermosensitive CLCs for wavelength-polarization dual-dimensional information encryption[43]. A multidimensional security label has also been designed using a photochemical dual-responsive CPL material that has wavelength, intensity, and chirality characteristics for information encoding[44]. Nevertheless, research on information encryption technology with multilevel security developed from CPL-active materials based on CLCs is still less reported, and the practical applications of this technology has encountered a bottleneck (i.e., in terms of multiple responses, precise local tuning, high $g_{lum}$ value, and easy operation) due to the lack of relevant advanced

[1]Key Laboratory of Carbon Fibers and Functional Polymers, Ministry of Education; College of Materials Science and Engineering, Beijing University of Chemical Technology, Beijing 100029, China. [2]Institute of Advanced Materials and School of Chemistry and Chemical Engineering, Southeast University, Nanjing 211189, China. [3]Advanced Materials and Liquid Crystal Institute and Materials Science Graduate Program, Kent State University, Kent, OH 44242, USA. ✉e-mail: guojb@mail.buct.edu.cn; quanli3273@gmail.com

materials. Thus, constructing phototunable CLC systems with fluorochromic properties and good processing performance is highly desirable.

Herein, we provide a proof-of-concept for 4D information encryption using phototunable liquid crystal photonic capsules (LCPCs) with full-color CPL (Fig. 1). The RGB fluorophores in the helical superstructures of CLCs undergo light-driven Förster resonance energy transfer (FRET), which enables the phototuning of trichrome CPL and fluorescence. Benefiting from the encapsulated luminescent CLCs in the microcapsules, the LCPCs feature both color tunability and easy processability, and these LCPCs are expected to be used as building blocks of solid materials for phototunable full-color CPL. With increasing UV irradiation time, the fluorescence of the single component LCPC (LCPC-S) remains unchanged, whereas the fluorescent intensities of RGB luminogens in binary or ternary LCPCs (LCPC-D or LCPC-T) exhibit strong time dependence, which can be attributed to the different FRET efficiencies. Notably, the fluorescence at any intermediate state is stable when stored in the dark because the fluorescent molecules undergo no thermal relaxation. Full-color CPL can be facilely achieved by mixing different LCPCs, and the RGB value can be regarded as a function of irradiation time and LCPC components. On the basis of the full-color CPL and time response properties, a 4D bar code is designed and decomposed into four characteristics: monochrome fluorescence, time dependent fluorescence/CPL, full-color emission, and different polarization states (fluorescence/CPL). These results pave the way for advanced data storage encryption media and photonic devices.

## Results and discussion
### Dynamic FRET systems

To realize chirality transfer from the chiral environment to achiral fluorescent molecules, a FRET platform has been constructed in CLCs as a brilliant chiroptical medium. The chiral fluorescent molecule (S)-CNB is selected as a chiral dopant and an energy donor, and the achiral molecular switches DG and DR serve as acceptors, with a significant absorption shift from their open- to closed-isomers. They are successfully synthesized (for details, see the Supplementary Method)[45–47], and their chemical structures are confirmed by nuclear magnetic resonance (NMR) spectra and high-resolution mass spectra (HR-MS) (Supplementary Figs. 1–14). Furthermore, (R)-CNB with the opposite handedness is also synthesized and served as a contrast molecule (Supplementary Figs. 15–18). The chiral induction ability of (S)-CNB can be measured by the Grandjean-Cano method (Supplementary Fig. 19), and the value of its helical twisting power (HTP) in the 5CB host is calculated as 11.43 μm$^{-1}$. (S)-CNB can effectively induce nematic LCs (5CB) to form helical superstructures, as verified by a pair of mirror circular dichroism (CD) spectra showing a negative Cotton effect for (S)-CNB and positive signals for (R)-CNB. (Supplementary Fig. 20b). (S)-CNB emits blue fluorescence at 436 nm in the 5CB host and shows a high fluorescence quantum yield of 32.46%, accompanied by a strong right-handed CPL with a $g_{lum}$ value of up to −0.33 (Fig. 2a and Supplementary Fig. 21). (R)-CNB in the 5CB host exhibits the opposite CPL signals, of which the $g_{lum}$ value can reach +0.38 (Supplementary Fig. 21). The molecular switches DG and DR can undergo reversible photocyclization/cycloreversion under UV and visible light irradiation

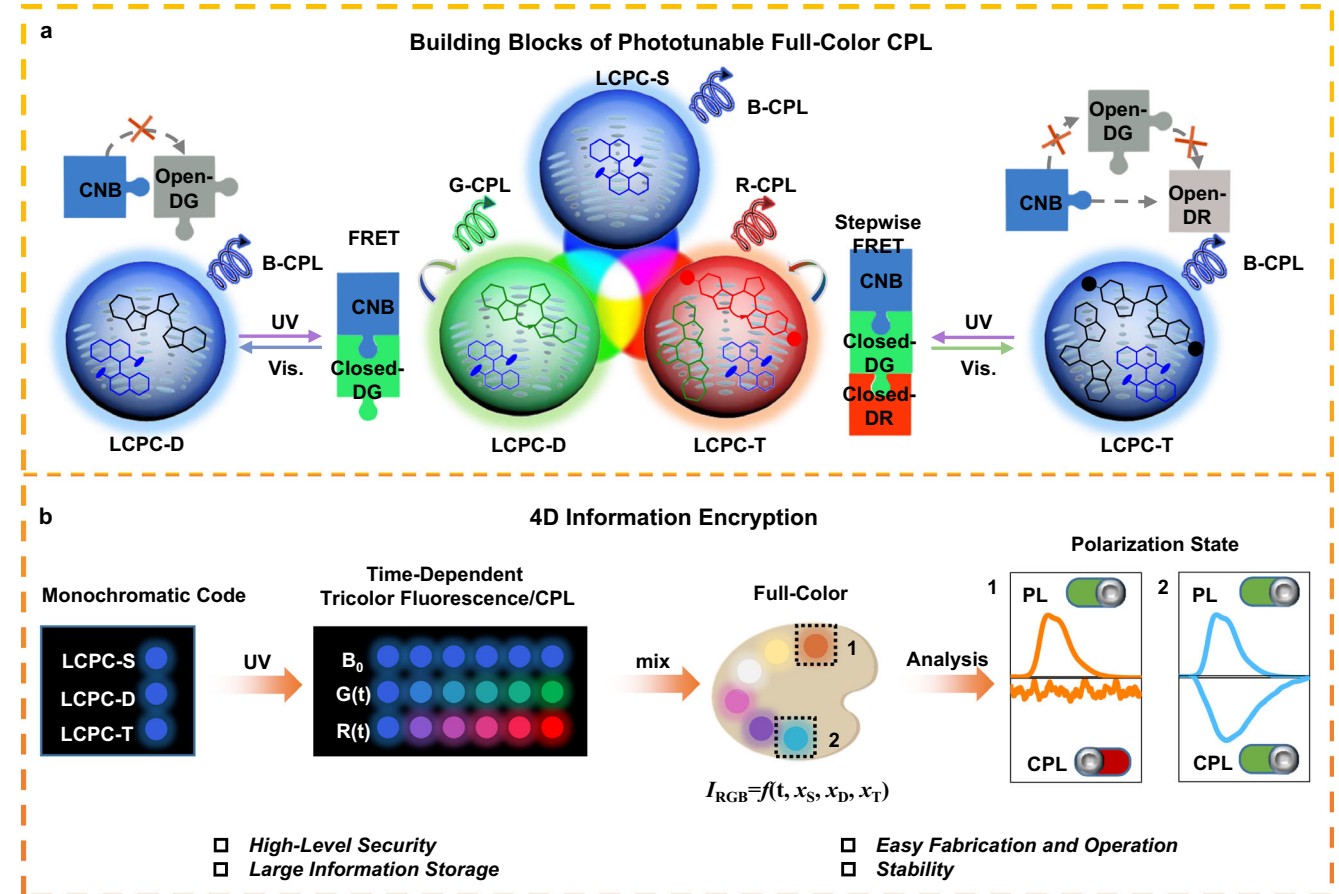

**Fig. 1 | Schematic illustration of phototunable liquid crystal photonic capsules (LCPCs) with full-color circularly polarized luminescence (CPL) and their application in four-dimensional (4D) information encryption. a** Light-driven Förster resonance energy transfer (FRET) processes of different component LCPCs used as building blocks for phototunable full-color CPL. **b** Principle design of 4D information encryption consisting of fluorescence, time response, full-color and CPL characteristics.

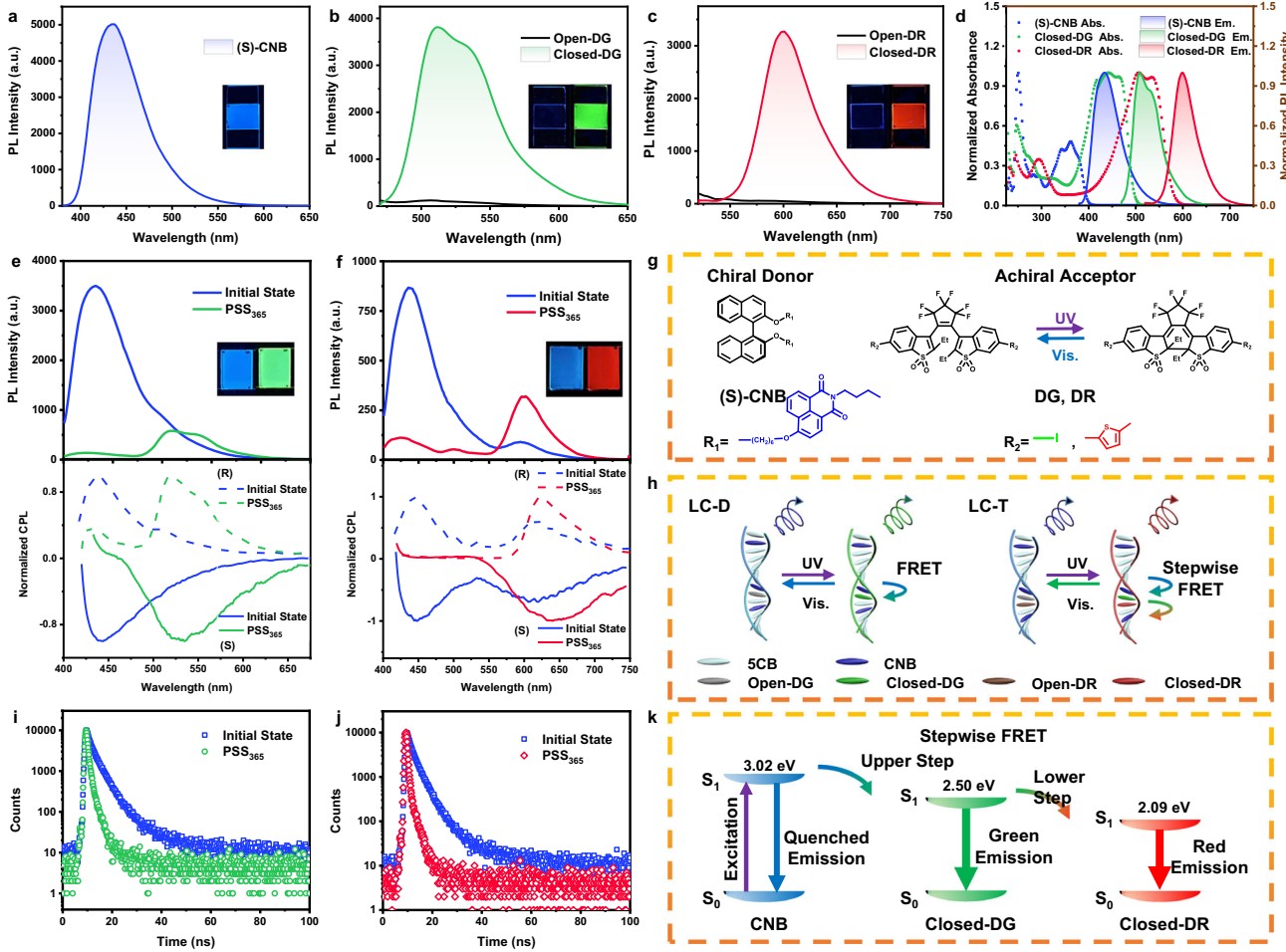

**Fig. 2 | Dynamic FRET systems constructed in the helical superstructures of CLCs. a** Fluorescence spectrum of (S)-CNB in the 5CB host (3.0 wt%, $\lambda_{ex}$ = 365 nm); the inset shows the fluorescent photograph of (S)-CNB. Fluorescence spectra of **b** DG (1.0 wt%, $\lambda_{ex}$ = 450 nm) and **c** DR (1.0 wt%, $\lambda_{ex}$ = 500 nm) in the 5CB host before and after UV irradiation (1.0 mW/cm²); the insets show the fluorescent photographs (black lines for open-isomers and green/red lines for closed-isomers). **d** Normalized absorption and normalized fluorescence spectra of (S)-CNB, closed-DG, and closed-DR (blue, green, and red dotted lines for the absorption of (S)-CNB, closed-DG, and closed-DR, blue, green, and red solid lines for the fluorescence of (S)-CNB, closed-DG, and closed-DR). Fluorescence and mirror CPL spectra of **e** LC-D (CNB:DG = 3.0 wt% :1.5 wt%, $\lambda_{ex}$ = 365 nm) and **f** LC-T (CNB:DG:

DR = 3.0 wt% :1.5 wt% :1 wt%, $\lambda_{ex}$ = 365 nm) before and after UV irradiation (1.0 mW/cm²); the insets show the fluorescent photographs (dotted lines for (R)-LC systems and solid lines for (S)-LC systems). **g** Chemical structures of chiral donor (S)-CNB and achiral acceptors DG and DR. **h** Schematic illustration of phototunable CPL and fluorescnece in LC-D and LC-T through dynamic FRET processes. Fluorescence decay profile of (S)-CNB in **i** (S)-LC-D and **j** (S)-LC-T before and after UV irradiation ($\lambda_{ex}$ = 365 nm, monitored at 434 nm) (blue squares for initial states of (S)-LC-D and (S)-LC-T, green circle for PSS$_{365}$ of (S)-LC-D, red rhombus for PSS$_{365}$ of (S)-LC-T). **k** Schematic diagram of stepwise FRET from CNB to closed-DR in the LC-T system in which closed-DG serves as the intermediate bridge.

(Fig. 2g and Supplementary Figs. 22 and 23), and exhibit instantaneous "turn-on"-type fluorescence in the 5CB host under UV irradiation (Fig. 2b, c), with fluorescence quantum yields of 20.64% and 16.25%, respectively. To further investigate the photocyclization/cycloreversion processes of DG and DR, their molecular structures in different photostationary states are tracked using NMR spectra. New signals are observed in the ¹H-NMR and ¹⁹F-NMR spectra of DG in PSS$_{365}$, which can be ascribed to the generation of closed-isomers (Supplementary Figs. 24 and 25). Based on the quantitative analysis of the related integral peaks, the ratio of open-DG to closed-DG is 12%:88% in PSS$_{365}$, and the photocycloreversion conversion is almost 100% under light irradiation at 450 nm. After aryl substitution, the photocyclization efficiency of DR can reach 100%, that is, the open-isomer can completely transform into closed-DR (Supplementary Figs. 26 and 27). Unfortunately, the cycloreversion conversion rate of DR is only 18.8% under light irradiation at 500 nm after a long time (10 h), which corresponds with the extremely low cycloreversion quantum yield of DR ($\Phi_{C \to O} < 10^{-5}$) reported in the literature[47]. It has been found that the

electron-donating effect of 2-methylthiophene hinders the isomerization of DR from the closed-state to the open-state[48–51].

The absorption bands of open-DG and open-DR are located in the UV region. After isomerization to their closed-forms, the absorption peaks dramatically redshift to the visible region and largely overlap with the peaks in the emission spectra of (S)-CNB and closed-DG, respectively (Fig. 2d). It is well known that FRET efficiency strongly depends on the spectral overlapping area and the distance between the donor and acceptor (i.e., the concentration of acceptor)[52–54]. The ideal spectral features of DG and DR enable phototunable FRET processes. Binary ((S)-CNB/DG, (S)-LC-D) and ternary ((S)-CNB/DG/DR, (S)-LC-T) FRET systems are built in the helical superstructures of CLCs (Fig. 2h). Initially, the helical superstructures generate blue CPL from the donor (S)-CNB. When exposed to UV light, FRET from (S)-CNB to closed-DG or stepwise FRET[55,56] from (S)-CNB to closed-DR occurs because of the photocyclization of DG and DR, leading to green or red CPL emission. As anticipated, the fluorescence peak of the (S)-LC-D system shifts from 434 nm to 510 nm under UV irradiation, with a

remarkable color migration from blue to green (Fig. 2e). The CD and CPL spectra of the (S)-LC-D system reveal the synergetic effects of energy and chirality transfer through the FRET process[24,26,57,58]. The edge of CD absorption broadens from 395 nm to 459 nm under UV irradiation, corresponding to the absorption band of closed-DG (Supplementary Fig. 28b). The right-handed CPL signal shifts from 450 nm to 530 nm, with the $g_{lum}$ value decreasing from −0.21 to −0.10, indicating that the emission center is transferred from the CPL emitter (S)-CNB to the achiral fluorophore DG (Fig. 2e and Supplementary Fig. 29). The synergistic transfer of energy and chirality can also be realized by changing the chiral donor from (S)-CNB to (R)-CNB. The (R)-LC-D system exhibits left-handed CPL photoswitching from blue to green emission under UV illumination, which causes the $g_{lum}$ values to change from +0.17 to +0.14 (Supplementary Figs. 28a and 29). The radiative decay of (S)-LC-D is quantified by time-resolved fluorescence lifetime measurements. In the absence of DG, the lifetime of (S)-CNB is measured as 4.601 ns (Supplementary Fig. 30 and Supplementary Table 1). Under UV irradiation, the lifetime of (S)-CNB in the (S)-LC-D system decreases to 0.860 ns, and the FRET efficiency is calculated as 81.3% (Fig. 2i and Supplementary Table 1). For the (S)-LC-T system, (S)-CNB serves as a chiral donor for both closed-DG and closed-DR, and closed-DG is employed as an intermediate bridge within a FRET cascade from (S)-CNB to closed-DR (Fig. 2k). To verify the cooperation of energy and chiral transfer in the stepwise FRET process, the (R)-LC-T system is prepared for a parallel experiment. Under UV irradiation, the blue fluorescence and CPL ($g_{lum}$ = −0.13 for (S)-LC-T and $g_{lum}$ = +0.084 for (R)-LC-T) are quenched entirely, and red fluorescence and CPL ($g_{lum}$ = −0.15 for (S)-LC-T and $g_{lum}$ = +0.21 for (R)-LC-T) are located at approximately 600 nm, far away from the emission peak of (S)-CNB or (R)-CNB (Fig. 2f, Supplementary Figs. 31a and 32). It is worth noting that both (S)-LC-T and (R)-LC-T systems in the initial state show CPL signals at around 600 nm because DR is so photosensitive that partial photocyclization is induced by UV-excited light during the measurement. To theoretically prove that (S)-LC-T and (R)-LC-T systems only generate blue CPL at the initial state, CPL testing is carried out using excited light with different wavelengths. When excited with visible light (430 nm and 500 nm), the systems have no CPL signal at 600 nm because the photocyclization of DR cannot be triggered (Supplementary Fig. 33). The mirror CD spectra of (S)-LC-T and (R)-LC-T in different states also confirm the chiral transfer (Supplementary Fig. 31b). In the photostationary state of 365 nm ($PSS_{365}$), the lifetime of (S)-CNB decreases to 0.278 ns, and the FRET efficiency is calculated as 94.0% (Fig. 2j and Supplementary Table 2). For comparison, a sample containing only (S)-CNB and DR ((S)-LC-B/R) is prepared, and it possesses a much lower FRET efficiency of 67.0% without the assistance of the DG intermediate bridge (Supplementary Fig. 34 and Supplementary Table 2). This stark contrast suggests that DG plays an indispensable role in the stepwise FRET process. By introducing the lower step of FRET, the efficiency of the upper step from (S)-CNB to closed-DG can be enhanced, increasing by 12.7%[59]. In addition, the lifetime of DG decreases from 1.229 ns to 0.584 ns after DR doping, and the FRET efficiency of the lower step from closed-DG to closed-DR is 52.5% (Supplementary Fig. 35 and Supplementary Table 3).

## Phototunable LCPCs

Small molecular CLCs with liquidity usually require sandwich devices (i.e., LC cells), which limits their practical applications. To address this issue, we present a facile and satisfactory method for encapsulating CLCs into microcapsules, which are capable of phototunable CPL and have excellent processing ability. The optical properties of (R)-LC systems are roughly the same as those of (S)-LC systems, except for the opposite handedness of CPL signals. Therefore, a chiral donor with left-handedness is chosen to further investigate the phototunable performance in LCPCs. Three types of LCPCs are fabricated by well-established interfacial polymerization[60]: single component (S)-CNB

(LCPC-S), a binary FRET system (LCPC-D), and a ternary FRET system (LCPC-T) (Supplementary Fig. 36). As observed by polarizing optical microscopy (POM), these LCPCs all have good spherical morphologies and particle diameters in the range of 10-15 μm (Supplementary Fig. 37). LCPCs appear iridescent under the crossed polarizers due to the birefringence of the liquid crystals. For example, the phase transition of LCPC-D is observed at a heating rate of 1 °C/min, and the transition from the cholesteric phase to an isotropic state occurs at 28 °C (Supplementary Fig. 38). LCPCs possess outstanding thermal stability and their spherical morphology can be maintained well at 100 °C. When cooled to room temperature, LCPCs revert back to the cholesteric phase. LCPCs show smooth shells with a polymer wall thickness of 500 nm when observed by scanning electron microscopy (SEM), and this thin shell endows LCPCs with excellent chromogenic performance (Fig. 3a, b). By adding LCPCs to an 8 wt% polyvinyl alcohol (PVA) aqueous solution and spin-coating the solution onto a glass substrate, the LCPC film is fabricated with a thickness of 22.75 μm and some randomly distributed holes of approximately 10 μm (Fig. 3c). The LCPC films also exhibit good phototunable fluorescence and CPL performance because the fluid cores provide sufficient space for the photocyclization of DG and DR. The changes in their fluorescence with UV irradiation time (t) are systematically characterized. The blue fluorescence of LCPC-S remains almost unchanged (Fig. 3e). LCPC-D and LCPC-T show the strongest emission in the blue band when t = 0; as t increases, the fluorescence in the blue band gradually decreases, and new emission emerges in the green band and red band, indicating that effective FRET channels are established under UV irradiation (Fig. 3f, g). The fluorescent images of LCPC films can give a more direct view of the time-dependent emission color changes under UV irradiation, and the diversification of fluorescent color can be clearly seen (Fig. 3h). By drawing the points corresponding to their fluorescent spectra on the CIE (1931) chromaticity diagram, the coordinates of LCPC-S, LCPC-D, and LCPC-T films in $PSS_{365}$ are found to be located in the blue, green, and red regions, respectively (Fig. 3i). The LCPC-D film also shows good reversibility, almost reverting to its initial state under 450 nm blue light irradiation for 20 min, and its emission can be repeatedly switched between blue emission and green emission by alternating light irradiation with 365 and 450 nm for at least 10 cycles (Supplementary Fig. 39). The fluorescence spectra of the LCPC-T film reveal that the predominant single peak in the red band in $PSS_{365}$ transforms into a doublet of peaks located at 443 nm and 600 nm under light irradiation at 500 nm for sufficient time, leading to emission color migration from red to pink (Supplementary Fig. 40). These LCPC films in $PSS_{365}$ have good thermal stability and can be stored at room temperature for one week without conspicuous degradation (Supplementary Fig. 41). Interestingly, LCPCs can also generate intense CPL, and the RGB tricolor CPL in $PSS_{365}$ can be attributed to LCPC-S ($\lambda_{em}$ = 445 nm, $g_{lum}$ = −0.11), LCPC-D ($\lambda_{em}$ = 535 nm, $g_{lum}$ = −0.08), and LCPC-T ($\lambda_{em}$ = 639 nm, $g_{lum}$ = −0.063) (Fig. 3j and Supplementary Fig. 42). Notably, the $g_{lum}$ values of CPL originating from LCPCs are slightly lower than those from LC cells, and this phenomenon can be attributed to the imperfect planar orientation of the helical superstructures in the microcapsules (Fig. 3d)[61–63].

## Full-color CPL and fluorescence

LCPCs with RGB tricolor CPL and fluorescence are capable of achieving full-color emission according to additive color theory[64–66]. The microcapsules can protect CLCs from the external environment and prevent unnecessary FRET between them. When exploring full-color emission, tricolor LCPCs serve as the building blocks, and the colors of films are ascribed to the superposition of individual LCPCs rather than energy transfer. In addition, LCPC-D and LCPC-T feature photoinduced fluorochromism, which imparts phototunability to full-color systems. The representative blend colors, i.e., cyan, yellow, purple, and white, are designed based on the CIE coordinates (Supplementary Table 4).

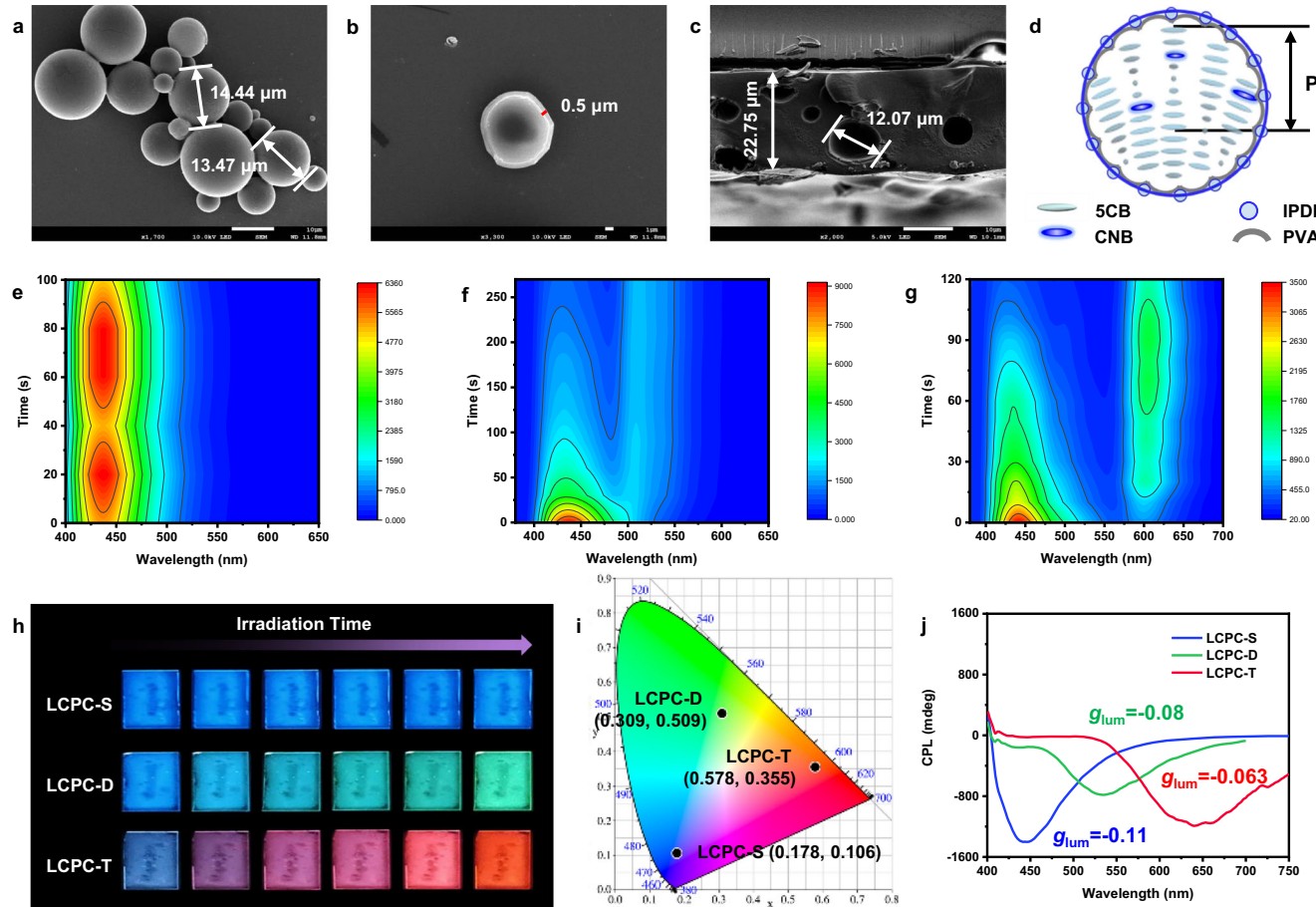

**Fig. 3 | Characterization of phototunable LCPCs and their time-dependent fluorescence. a** SEM images of the morphology of LCPC-D and **b** thickness of the polymer shell (The scale bars are 10 μm and 1 μm). **c** Cross-sectional image of the LCPC film (The scale bar is 10 μm). **d** Orientation of helical superstructures in microcapsules. Contour maps of **e** LCPC-S, **f** LCPC-D, and **g** LCPC-T, showing the changes in fluorescent intensity with UV irradiation time ($\lambda_{ex}$ = 365 nm, 1.0 mW/cm$^2$).

The color gradient represents the fluorescence intensity, with red for high value and blue for none emission. **h** Fluorescent images of LCPCs with increasing of UV irradiation time (1.0 mW/cm$^2$). **i** CIE chromaticity diagram of LCPC films in PSS$_{365}$. **j** CPL spectra of LCPCs in PSS$_{365}$ ($\lambda_{ex}$ = 365 nm) (blue line for LCPC-S, green line for LCPC-D in PSS$_{365}$, and red line for LCPC-T in PSS$_{365}$).

They all emit blue fluorescence in the initial state, with a blue peak dominating the whole spectrum (Fig. 4a–d). Under UV irradiation, the emission of LCPC-D (T) changes from blue to green (red) due to the photoinduced FRET process, and the films show an additive color with a doublet or triplet peaks located at the corresponding wavelengths of LCPC-S, LCPC-D, and LCPC-T. By blending different components of trichroic LCPCs, photoswitchable blue-to-multicolor films are prepared (Fig. 4e and Supplementary Figs. 43–46). The coordinates of LCPC-S, LCPC-D, and LCPC-T on the CIE (1931) chromaticity diagram present three vertices of a triangle, and the fine-tuning of emission color can be realized along the edges of the triangle by changing the blending ratio (Fig. 4f, g). Notably, the CIE coordinate of the S/D/T film in PSS$_{365}$ is estimated to be $(x, y)$ = (0.301, 0.319), which is close to that of pure white light (0.333, 0.333)[67]. These films with diverse LCPC components also exhibit full-color CPL signals, accompanied by $g_{lum}$ values in the range of −0.063 – −0.11 (Fig. 4h). LCPCs have outstanding processability, and in addition to being processed on a glass substrate, they can be easily processed on flexible substrates, such as polyethylene terephthalate (PET) film. As shown in Fig. 4i, LCPC-S, LCPC-D, and LCPC-T are overlaid on the same PET film through the bar-coating method. The initial film only emits blue fluorescence, when exposed to UV light, a rainbow film with simultaneous blue, green, and red luminescence is obtained. This film can bend at large angles, suggesting that it has good flexibility. LCPCs can also be used to program patterns on flexible films by direct ink writing. A dipping pen with LCPC solution

as the ink is used to write interesting patterns on the PET film. Before UV irradiation, the letters written by different LCPC inks all emit blue fluorescence; while after illumination, a colored "LCM" is observed due to the photoinduced FRET process (Fig. 4j). This photoswitchable ink can be reset to its initial state when exposed to visible light with 500 nm and 450 nm for a long time (Supplementary Fig. 47). Additionally, the LCPC films featuring photoswitchable colors can be applied in flexible displays. Under exposure to UV light using a photomask, a preprogrammed pattern, a red lotus, is clearly imprinted on the blue background (Fig. 4k).

**Multilevel information encryption**

Multifunctional LCPC films have integrated fluorescence, CPL, full-color emission, and time response characteristics, showing great superiority in the field of information encryption. First, 2D information encryption based on fluorescence and full-color emission is demonstrated (Fig. 5a). The binary codes of the standard 8-bit ASCII characters are programmed using four types of LCPCs, including LCPC-S, LCPC-D, LCPC-T, and LCPC-S/D/T. These four LCPCs all exhibit blue fluorescence and are indistinguishable from each other. If the information is read directly, a meaningless combination of the letters "GDBQ" is obtained. In the decryption step, UV light and color serve as the two crucial keys. LCPCs shed their camouflage of blue fluorescence under UV irradiation and show their real colors (blue, green, red, and white). However, the colored code still represents fake information,

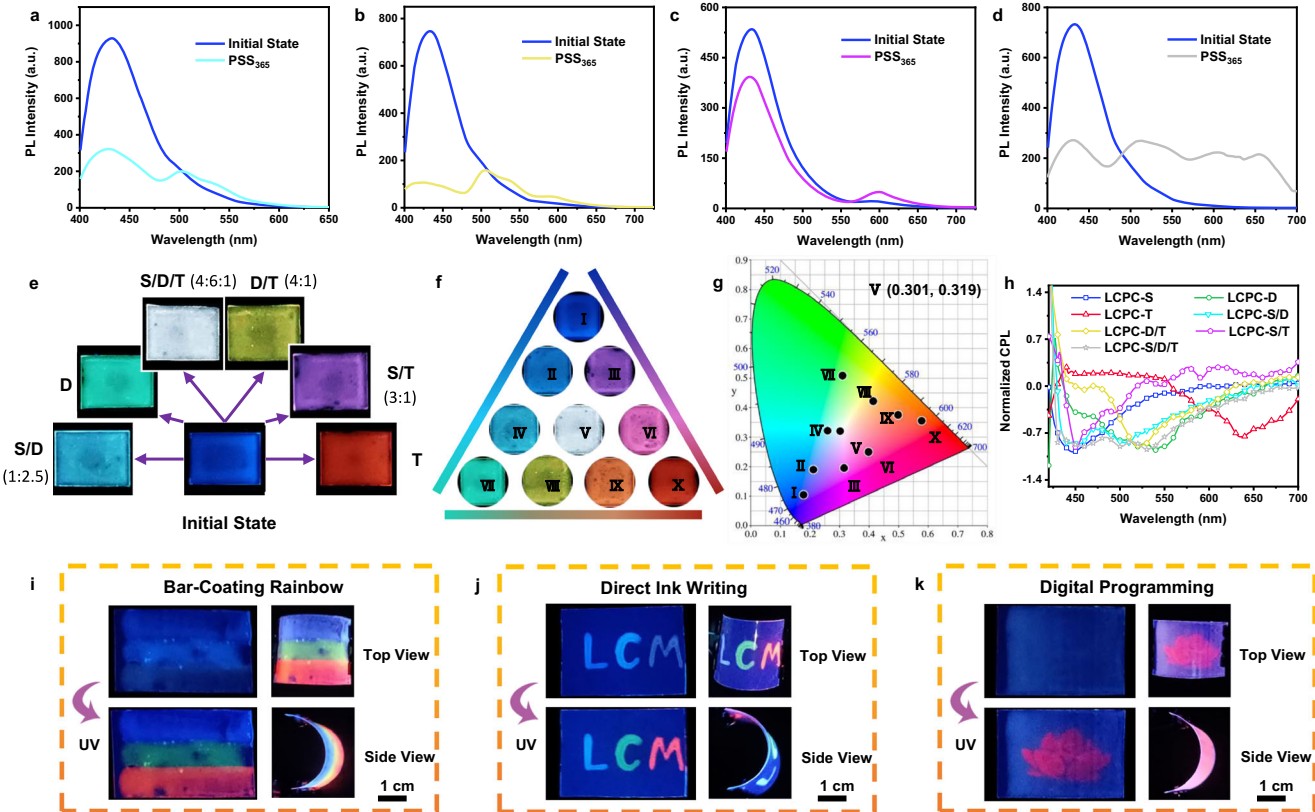

**Fig. 4 | LCPC films with full-color emission and their multiple photonic applications.** Fluorescence spectra of **a** LCPC-S/D (blue line for initial state and cyan line for PSS₃₆₅), **b** LCPC-D/T (blue line for initial state and yellow line for PSS₃₆₅), **c** LCPC-S/T (blue line for initial state and purple line for PSS₃₆₅), and **d** LCPC-S/D/T films (blue line for initial state and silver gray line for PSS₃₆₅) in different states (λₑₓ = 365 nm). **e** Fluorescent images of LCPC films showing the emission color changes under UV irradiation. **f** LCPC films with full-color emission under UV irradiation by blending different ratios of trichroic LCPCs. **g** CIE chromaticity diagram of full-color LCPC films in PSS₃₆₅. **h** CPL spectra of full-color LCPC films in PSS₃₆₅ (λₑₓ = 365 nm) (blue line for LCPC-S, green line for LCPC-D, red line for LCPC-T, cyan line for LCPC-S/D, yellow line for LCPC-D/T, purple line for LCPC-S/T, and silver gray line for LCPC-S/D/T). Fluorescent images of **i** a rainbow film fabricated by the bar-coating method, **j** colored letters fabricated by direct ink writing, and **k** digital programming facilitated by photomasks (The scale bars are 1 cm).

and further decryption of the true information requires the second key color. Each line has 2 numbers for every color, and with the color as the classification standard, the numbers of the same color are selected from each row and combined again in order to obtain a new code. After color rearrangement, the true information "FRET" is successfully read. Subsequently, 3D information encryption is designed by introducing CPL into 2D encryption technology (Fig. 5b). To realize a geminate label carrying distinct information, four kinds of LCPCs are prepared and designated LCPC-G, LCPC-D, LCPC-R, and LCPC-T. Among them, LCPC-G and LCPC-R only encapsulate molecular switches DG and DR, which emit fluorescence solely under UV irradiation. In stark contrast, LCPC-D and LCPC-T generate both fluorescence and CPL signals due to chirality transfer through the FRET process. Based on the differences in emission behaviors, these four types of LCPCs are arranged in an array of 30 × 9 pixels according to the predefined program. Under UV illumination, the wrong message composed of colored letters "BLUE" can be directly read by the naked eye. However, the true information hidden in the deep layer needs to be identified by analytical tools and exported. When the array is scanned using the analysis tools, only the parts of the letters consisting of LCPC-D and LCPC-T generate CPL signals. Thus, the CPL information "LC" can be obtained by capturing CPL signals. In addition, visualized 3D information decryption is presented by introducing left-handed CLC reflective films to amplify the chirality of the LCPCs. Taking advantage of the selective reflection of CLC films, the gₗᵤₘ values of tricolor LCPCs are strongly enhanced to −1.58 for LCPC-S, −1.72 for LCPC-D, and −1.39 for LCPC-T (Supplementary Fig. 48). Left-handed CPL is prohibited from passing through the photonic band gap (PBG) of the CLC film, while CPL located at

other wavelengths is not affected (Supplementary Fig. 49a). Therefore, the CLC reflective films with different PBGs can act as deciphering keys when observed in the left-polarized window: the interference message can be filtered out by choosing the true decipher (Supplementary Fig. 49b). The real information can be directly read by the naked eye through UV irradiation, color rearrangement, and polarized filtration without the assistance of analytical tools (Supplementary Fig. 50).

In addition, by taking advantage of the phototuning and time response characteristics, a 4D bar code with a higher security level and more complicated decryption is crafted by six kinds of LCPC films (Fig. 6). The LCPC-S film is photostable, and the LCPC-D, LCPC-T, LCPC-D/T, and LCPC-S/D/T films change their colors from blue to green, red, yellow, and white under UV irradiation. Moreover, the intensities of the RGB components in these LCPC films show strong time dependence because the absorption spectra of DG and DR change along with the irradiation time, leading to a gradual enhancement in FRET efficiency. The fluorescence intensity $I(t)$ can be described as a function of irradiation time and RGB components. The intensity variations $I(t)/I_0$ and $I(t)/I$ are plotted over time, where $I_0$ is the intensity of CNB at $t = 0$ s, and $I$ is the intensity of DG or DR in PSS. LCPC films with different components have identifiable $I$-$t$ curves at the same irradiation time (e.g., $t = 30$ s), and each LCPC film shows a different rate of RGB fractional change (i.e., B/G or B/T). By storing these $I$-$t$ curves of LCPC films in the computer, they can be used as an important component of decryption. The designed bar code is composed of three groups of eight bars, corresponding to three groups of ASCII binary codes. When the RGB components of each bar are consistent with the prestorage data, the computer outputs the number "1"; otherwise, it outputs the

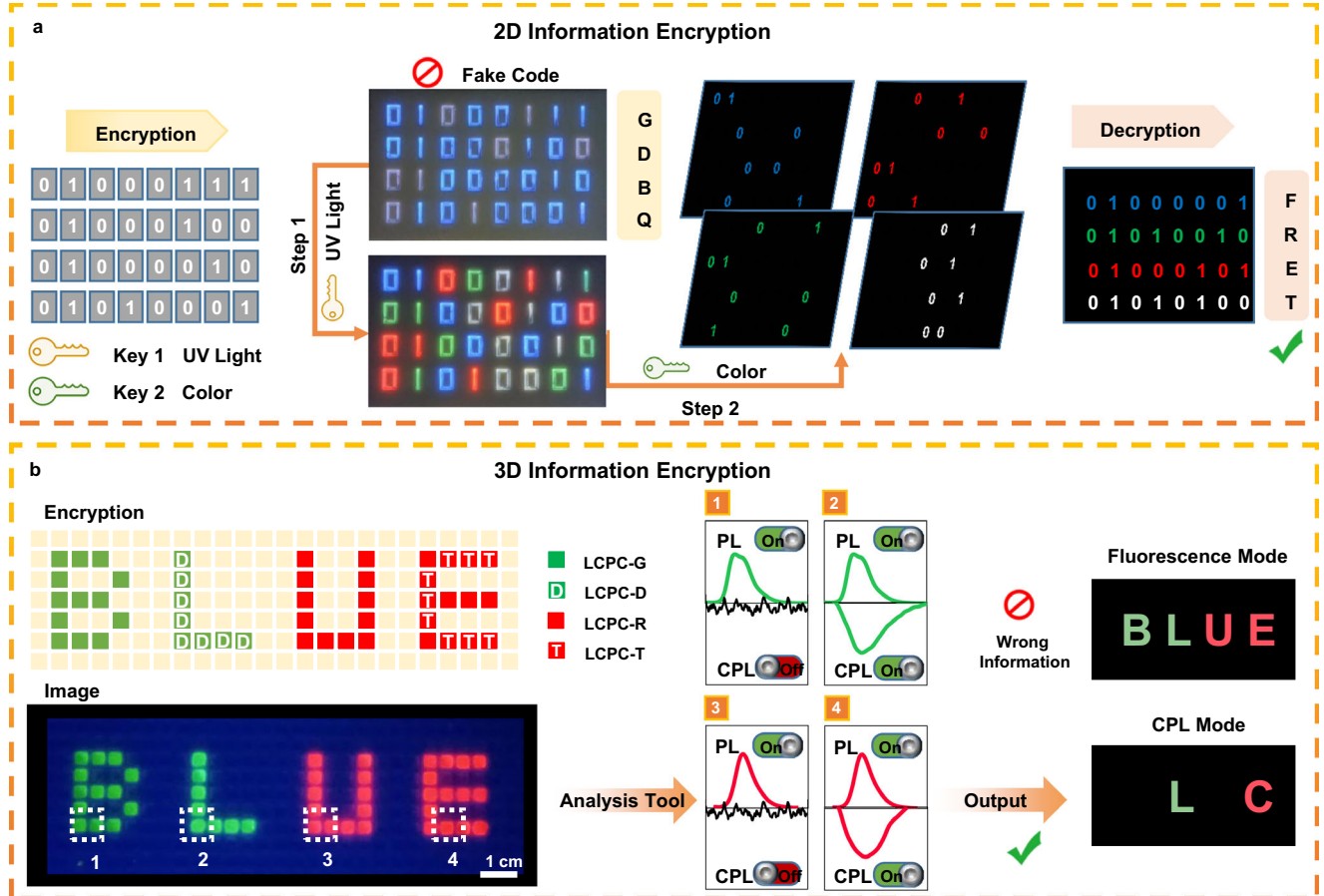

**Fig. 5 | Potential applications in information encryption using multifunctional LCPC films. a** Binary code constructed by LCPC films for 2D information encryption, with UV light and color as the decryption elements. **b** Demonstration of 3D information encryption using the LCPC array with different emission behaviors between fluorescence and CPL (The scare bar is 1 cm).

number "0". Furthermore, a fluorescent bar fabricated with LCPC-G, which emits no CPL signals, serves as camouflage in this 4D bar code. Based on the aforementioned aspects, the concept of this 4D bar code with its decryption process is demonstrated. The encryption algorithm is described as (color channel number, irradiation time, polarization state). For instance, a cipher is defined as (5, 90 s, CPL), and the initial blue-emissive bar code cannot be recognized due to its monochromic channel (Fig. 6i). When exposed to UV light for 30 s, the 1D bar code transforms into a 2D colored bar code (Fig. 6ii). However, at this time node, the RGB components of the bar code ($T_1$) vary from the preinput ($T_2$), so the information cannot yet be read. Only if the specified irradiation time (90 s) is reached can the information "CTL" be output because the RGB components are consistent with the cipher (Fig. 6iii). The information obtained at this point is still false owing to the camouflage of the fluorescent code. Finally, the real information "CPL" is obtained through a CPL detector (Fig. 6iv). Such a 4D bar code is much more complex than the traditional bar code, because massive data would be carried on for decryption under the condition of the uncertain cipher.

In summary, we have developed device-friendly solid films with phototunable full-color CPL. By establishing FRET platforms among the chiral donor (S)-CNB and achiral molecular switches (DG, DR) in the helical superstructures of CLCs, photoswitchable trichromatic CPL emissions with a maximal $g_{lum}$ value of up to −0.21 are generated through the cooperation of energy and chiral transfer. LCPCs encapsulating luminescent CLCs are prepared by interface polymerization, where the liquid cores ensure excellent phototunable performance and the polymer shells impart good processability. These LCPCs can exhibit RGB tricolor CPL signals with $g_{lum}$ values ranging from −0.063 ~ −0.11 and show strong time dependence due to the enhanced FRET efficiency along with the irradiation time. By blending tricolor LCPCs into the PVA medium with different mass ratios, LCPC films with phototunable full-color emission are fabricated. On the basis of these phototunable fluorochromic properties, time-responsive behavior, and different polarization states, the concept of 4D data encryption and decryption is demonstrated. Such functional LCPC films with integrated fluorescence, time response, full-color, and CPL characteristics will be of great interest for their potential applications in multi-level information encryption.

## Methods
### Materials
All the solvents and reagents are obtained from commercial sources and used as purchased without further purification unless otherwise noted. Nematic liquid crystal 5CB (99%) is purchased from Jiangsu Hecheng Display Technology Co., Ltd (HCCH). (S)−1,1'-binaphthol (98%) and (R)−1,1'-binaphthol (99.87%) are purchased from Bide Pharmatech (China) Co., Ltd., and used as received. 4-Bromo-1,8-naphthalic anhydride (98%) is purchased from Beijing Ouhe Technology Co., Ltd., and used as received. Tetrahydrofuran (99.5%) is dried with 3 Å molecular sieves for more than 48 h before use. (S)/(R)-CNB, DG, and DR are synthesized and purified by column chromatography, and confirmed the molecular structures by ¹H-NMR, ¹³C-NMR, ¹⁹F-NMR, and HR-MS. The detailed synthetic procedures are shown in Supplementary Method. Column chromatography is carried out on silica gel (200−300 mesh).

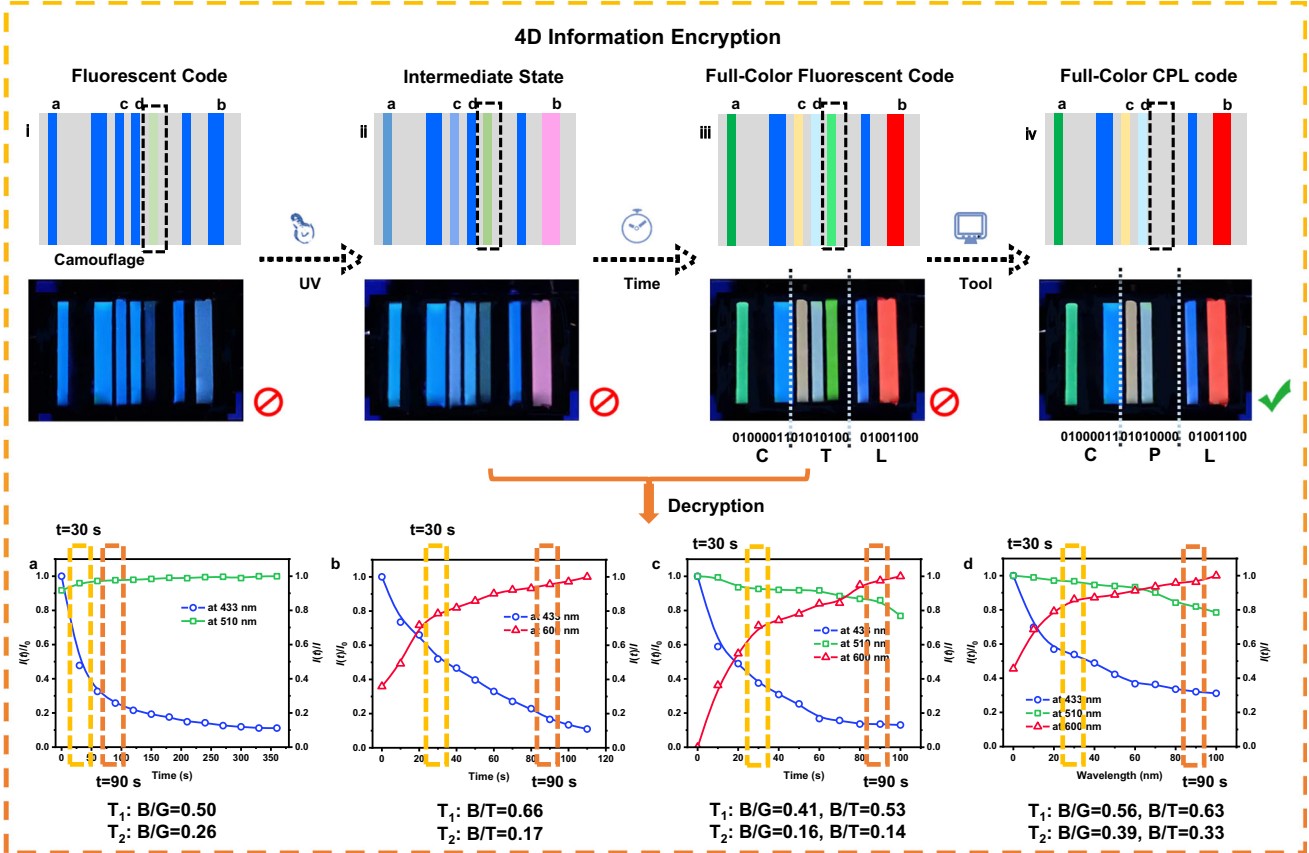

**Fig. 6 | Design and concept of 4D information encryption and decryption.** i-iv Dynamic changes from a 1D monochromatic bar code to a 4D colored bar code. *I*-t curves of RGB components in **a** LCPC-D (blue line for intensity at 433 nm and green line for intensity at 510 nm), **b** LCPC-T (blue line for intensity at 433 nm and red line for intensity at 600 nm), **c** LCPC-D/T (blue line for intensity at 433 nm, green line for intensity at 510 nm, and red line for intensity at 600 nm), and **d** LCPC-S/D/T films (blue line for intensity at 433 nm, green line for intensity at 510 nm, and red line for intensity at 600 nm) used for authenticating information.

## Preparation of LC systems

The LC cell is fabricated by two glass substrates spin-coated with 3.5 wt % PVA aqueous solution, and the PVA layers have predefined antiparallel rubbing directions to induce homogeneous alignment of CLC molecules. The CLC mixture consisting of 3.0 wt% (S)-CNB, 1.5 wt% DG, and 95.5 wt% 5CB is injected into the as-prepared LC cell with 10 μm spacers through the capillary effect. Then the (S)-LC-D system is fabricated. The (S)-LC-T system is prepared using the same process, except the CLC mixture is composed of 3.0 wt% (S)-CNB, 1.5 wt% DG, 1.0 wt% DR, and 94.5 wt% 5CB.

## Preparation of LCPCs

LCPCs are fabricated according to the previous work[60]. 0.96 g CLC mixture and 0.24 g IPDI are dissolved in 2 g dichloromethane as the oil phase, where the mass ratio of core and shell is 4:1. Then 4 g 8 wt% PVA aqueous solution and 8.8 mL $H_2O$ are added into the solution as aqueous phase, of which the content of PVA is 2 wt%. After stirred violently at 1700 r/min for 10 min, the emulsion is polymerized at 80 °C for 6 h with 2 or 3 drops of DBTDL. The obtained suspension is centrifuged at 159 × g for 5 min, and the precipitates are sonicated and dispersed in deionized water. The LCPC aqueous solutions are carried out for fluorescence and SEM characterizations. A part of the precipitates are freeze-dried to prepare the LCPC films.

## Preparation of LCPC films

The freeze-dried LCPCs are doped into 8 wt% PVA aqueous solutions and the concentration of LCPCs is 1 wt%. Then the mixture is sonicated for 1 h to disperse evenly. The PVA solution of LCPCs is spin-coated on

the glass substrate at a low speed or bar-coated on the PET substrate, and the substrates are placed on the hot-stage at 50 °C. After drying, the LCPC films are obtained.

## Spectroscopic measurements

Fluorescence spectra are carried out using a Hitachi F-4500 instrument. CD measurement is conducted on a JASCO 810 spectropolarimeter. CPL spectra together with $g_{lum}$ values are measured and recorded at room temperature on JASCO CPL-200. Fluorescence lifetimes are measured by a transient steady-state fluorescence instrument FLS980 using a time-correlated single photon counting (TC-SPC) setup. Corrected CIE coordinates and absolute fluorescence quantum yields are determined by FLS980.

## Calculation of energy levels

The energy levels corresponding to the first singlet excited states of (S)-CNB, DG, and DR are calculated by Gaussian 09 based on time dependent-density functional theory (TD-DFT) at B3LYP level. 6-31G-/-d-/ is selected as the basis set for (S)-CNB, and Lanl2DZ is selected as the basis set for DG and DR. The used program package is Gaussian 09 Rev. B.01.

## Calculation of FRET efficiency

According to Eq. (1), the FRET efficiency is calculated from the fluorescence lifetime, where $\tau_{AD}$ and $\tau_D$ are the average lifetime of donor in the condition with and without acceptor.

$$E = 1 - \tau_{AD}/\tau_D \qquad (1)$$

The fluorescence lifetime decay curves are fitted as the double exponential functions shown in Eq. (2), and the average lifetimes are calculated by Eq. (3).

$$I(t) = I_0 + A_1 \exp(-t/\tau_1) + A2 \exp(-t/\tau_2) \qquad (2)$$

$$\tau_{av} = (A_1\tau_1^2 + A_2\tau_2^2)/(A_1\tau_1 + A_2\tau_2) \qquad (3)$$

## Data availability

The data generated in this study are provided in the Supplementary Information. Any other data are available from the corresponding authors upon request.

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

## Acknowledgements

The authors acknowledge the supports from National Natural Science Foundation of China (Grant Nos. 52073017 and 51773009), Jiangsu Innovation Team Program, and the Fundamental Research Funds for the Central Universities.

## Author contributions

J.G. and Q.L. designed the experiments and directed the research. S.L. performed the experiments. Y.T. and W.K. assisted in compound preparations and characterizations. H.K.B. assisted in the design of figures. S.L., Y.T., W.K., H.K.B. and J.G. analyzed the data. S.L., H.K.B., J.G., and Q.L. discussed the interpretation of results and wrote the paper. All authors discussed the results and commented on the manuscript.

## Competing interests

The authors declare no competing interests.
