## [Peer Review File · Nature Communications]

Photo-Triggered Full-Color Circularly Polarized Luminescence Based on Photonic Capsules for Multilevel Information EncryptionReviewers' Comments:

Reviewer #1:

Remarks to the Author:

This paper presents a systematic study of Photo-Triggered Full-Color Circularly Polarized Luminescence Based on Photonic Capsules for Multilevel Information Encryption. The major novelty of the work relies on a combination of full-color emission, time response, and phototunable CPL for multilevel data encryption. A label that carries entirely distinct information is interesting and useful in information storage and anti-counterfeiting technology.

But I have some concerns about this work, enumerated below:

1) The circularly polarized luminescence is important in this work. However, the Introduction lacks the comparative research progress of CPL-active materials in the field of information encryption, which highlights the application bottleneck proposed by the authors due to the lack of relevant advanced materials. Several references could be considered. (Nat. Commun. 2022, 13, 7841; Angew. Chem. 2022, 134, e202205633; Adv. Mater. 2021, 33, 2103329; Adv. Funct. Mater. 2022, 32, 2204487; Chem. Eng. J 2022, 450, 138390.)

2) Since (S)-1,1'-binphthophenol acts as a chiral transfer template, these liquid crystal photonic capsules should radiate right-handed circularly polarized luminescence (CPL). According to the indicative spiral arrows in Fig. 1a, LCPC-S and LCPC-T show right-handed CPL, while LCPC-D presents left-handed CPL. The possible reasons?

3) The experimental characterization section is missing from the manuscript. The reader cannot understand the reliability of the data. How the luminescence dissymmetry factors (g_{lum}) are obtained, whether it is tested by the CPL-300 spectrometer or by a self-built optical platform.

4) The author highlight in the Introduction that the CPL can afford more dimension of information. Luminescent cholesteric liquid crystals are a powerful candidate for generating tunable CPL with large g_{lum} . Although these films with diverse components of LCPCs exhibit full-color CPL signals, their g_{lum} values are only in the range of $-0.063 \sim -0.11$. In addition, appropriate characterization data are rarely shown in the text, such as a direct view of right-handed CPL information. These results could affect the role of right-handed CPL in information encryption.

5) There are many sentence and word errors in the manuscript.

6) Currently, the combination of fluorescent materials and chiral liquid crystals has been widely used to develop CPL-active materials for their further development of various functional applications (Nat. Photonics, 2022, 16, 174; J. Mater. Chem. C, 2021, 9, 12590; J. Phys. Chem. Lett. 2021, 12, 598; Nat. Commun. 2020, 11, 5659.). For instance, a g_{lum} value is up to 1.9 from perovskites in chiral liquid crystal-based soft helix devices for cryptology and anti-counterfeiting (Matter 2022, 5, 2319.). Overall, the advantages of the CPL-activated photonic capsules prepared by the technical route of this work for information storage are not particularly prominent. There are many combinations to choose from.

In summary, I cannot recommend the manuscript for publication in its current state.

Reviewer #2:

Remarks to the Author:

In the manuscript, the authors proposed a considerable 4D information encryption strategy based on liquid crystal photonic capsules (LCPCs), which realized the phototunable multicolor CPL through energy transfer and chiral transfer processes. Through mixing different component LCPCs, the full-color CPL films can be obtained, which can be successfully applied to multilevel information encryption. This study paves the way for advanced data storage encryption media and photonic devices. Anyway, I think that the present work is suitable for publication in Nature Communications after considering the following issues.

1. The authors need to conduct NMR experiments to investigate the photocyclization/cycloreversion

recovery process upon UV and visible light irradiation for DG and DR.

2. In this study, the CNB molecule was synthesized with (S)-1,1'-binaphthol and its right-handed photophysical properties were studied. And how about the (R)-1,1'-binaphthol, it is suggested that the authors discuss it.

3. In Fig. 2f, the blue line has a weak emission band near 600nm, the authors should retest the spectrum to eliminate misunderstanding.

4. On the page 7, for the sentence "of which the efficiency is increased by 12.7%", the authors need to provide the data sources in detail.

5. Reversible photocyclization/cyclization under UV and visible light irradiation was confirmed in solution, and it is suggested to discuss the optical reversibility of LCPCs films through spectral experiments.

6. In Figure 4e, a series of colors are obtained by mixing different components of three color LCPC. It is suggested that the author display the corresponding component proportion of each state to make the results more scientific and rigorous.

7. In Figure 13, "A combination of the letters" GDBQ "will be obtained". There is no corresponding graph for "GDBQ".

8. In Figure 5b, the concept of another dimension (polarization state) of information encryption is proposed, but it is not really realized. The author should emphasize the advantages of this work compared with previous reports.

9. In Figure 6, it is recommended to explain how the bar code corresponds to a digital code.

10. The following paper, including research on multicolor luminescence and its application, can be referred to: *Angew. Chem. Int. Ed.* 2021, 60, 27171–27177; *Adv. Sci.* 2022,9, 2201523.

Reviewer #3:

Remarks to the Author:

In my opinion, the article is not suitable for publication in *Nature Communications* for the following reasons:

- The work is not innovative enough: There are many similar articles in the literature in which fluorescent molecules are used to perform data encryption, including also the ref. 50 by the same authors of this article; the only difference is that in ref. 50 thermally reconfigurable luminescent materials are used.

- Used physical systems are cholesteric capsules similar to those realized by other research groups that are completely ignored. Just to name a few of them:

1) Donato, M. G., Hernandez, J., Mazzulla, A., Provenzano, C., Saija, R., Sayed, R., ... & Cipparrone, G. (2014). Polarization-dependent optomechanics mediated by chiral microresonators. *Nature communications*, 5(1), 1-7.

2) Y. Li, J. J.-Y. Suen, E. Prince, E. M. Larin, A. Klinkova, H. Thérien-Aubin, S. Zhu, B. Yang, A. S. Helmy, O. D. Lavrentovich, and E. Kumacheva (2016). "Colloidal cholesteric liquid crystal in spherical confinement," *Nat. Commun.* 7(1), 12520..

3) Petriashvili, G., Chanishvili, A., Zurabishvili, T., Chubinidze, K., Ponjavidze, N., De Santo, M. P., ... & Barberi, R. (2019). Temperature tunable omnidirectional lasing in liquid crystal blue phase microspheres. *OSA Continuum*, 2(11), 3337-3342, ISO 690

In particular, in the last one, the dependence of the cholesteric properties on temperature is used to realize a tunable laser.

- Finally, the authors completely neglect aspects related to thermal, optical and mechanical stability of

their systems. All these characteristics play a fundamental role in application oriented investigations of the proposed physical systems.

Re: manuscript NCOMMS-22-50154

We sincerely thank the respected reviewers for their valuable time and helpful comments. We have carefully revised our manuscript and Supporting Information by taking into account the respected reviewers' comments as appropriate.

Our point-by-point response to the reviewers' comments and the changes made in our revised manuscript with red color highlighted track change as follows.

Reviewer 1:

“This paper presents a systematic study of Photo-Triggered Full-Color Circularly Polarized Luminescence Based on Photonic Capsules for Multilevel Information Encryption. The major novelty of the work relies on a combination of full-color emission, time response, and phototunable CPL for multilevel data encryption. A label that carries entirely distinct information is interesting and useful in information storage and anti-counterfeiting technology.”

Our response: We sincerely thank the respected reviewer for your valuable time in assessing the manuscript, recognizing its contemporary significance and originality.

“But I have some concerns about this work, enumerated below:

1) The circularly polarized luminescence is important in this work. However, the Introduction lacks the comparative research progress of CPL-active materials in the field of information encryption, which highlights the application bottleneck proposed by the authors due to the lack of relevant advanced materials. Several references could be considered. (Nat. Commun. 2022, 13, 7841; Angew. Chem. 2022, 134, e202205633; Adv. Mater. 2021, 33, 2103329; Adv. Funct. Mater. 2022, 32, 2204487; Chem. Eng. J 2022, 450, 138390.)”

Our response: Per suggestion, we have rewritten the introduction that summarizes the recent progress of CPL-active materials in the field of information encryption. The references mentioned by the respected reviewer are all cited and discussed in our revised manuscript. Compared with those previous works, the time dimension is introduced into CPL materials (color changing with time), and a 4D information encryption is constructed. Meanwhile, the means of encrypting the information and the steps of decryption are explicitly demonstrated.

“2) Since (S)-1,1'-binphthophenol acts as a chiral transfer template, these liquid crystal photonic capsules should radiate right-handed circularly polarized luminescence (CPL). According to the indicative spiral arrows in Fig. 1a, LCPC-S and LCPC-T show right-handed CPL, while LCPC-D presents left-handed CPL. The possible reasons?”

Our response: We are sorry for the misunderstanding caused by the opposite direction of the CPL label. The original design is intended for beauty, rather than representing left-handed CPL. To avoid the possible confusion, we have corrected the direction of the CPL arrows of LCPC-D, making it consistent with LCPC-S and LCPC-T in our revised Fig. 1.

“3) The experimental characterization section is missing from the manuscript. The reader cannot understand the reliability of the data. How the luminescence dissymmetry factors (g_{lum}) are obtained, whether it is tested by the CPL-300 spectrometer or by a self-built optical platform.”

Our response: The g_{lum} values are obtained by a JASCO CPL-200

spectrofluoropolarimeter. We have added detailed experimental characterizations in the section of Methods in our revised manuscript. Furthermore, all corresponding g_{lum} spectra of CPL testing have been added in our revised Supporting Information to ensure their reliabilities as Supplementary Fig. 19, 27, 30 and 40.

“4) The author highlight in the Introduction that the CPL can afford more dimension of information. Luminescent cholesteric liquid crystals are a powerful candidate for generating tunable CPL with large g_{lum} . Although these films with diverse components of LCPCs exhibit full-color CPL signals, their g_{lum} values are only in the range of -0.063~-0.11. In addition, appropriate characterization data are rarely shown in the text, such as a direct view of right-handed CPL information. These results could affect the role of right-handed CPL in information encryption.”

Our response: As the reviewer pointed out, large g_{lum} value is important for CPL in real applications, the g_{lum} values of the CLC systems in this work are on the order of 10^{-1} , of which the order of magnitude is consistent with those reported in the references. Meanwhile, the CLC films with selective reflection could separate circularly polarized light and have been used to obtain CPL-active materials with ultrahigh g_{lum} values. We also added these data as demonstrated later. In addition, under consideration of the application applicability of multilevel information encryption, LCPCs featuring phototunable CPL, time-dependence, and outstanding processability are explored. In the second half of the manuscript, the huge application potential of LCPCs is demonstrated, such as processing on the flexible substrate by spin-coating or bar-coating method or acting as the pen ink for direct writing. Although some CPL materials limited in the planar LC cell have large g_{lum} values, they do not possess the same easy-processable property of LCPCs.

Per suggestion, we have added the corresponding discussion of the visualized 3D information encryption in our revised manuscript and the data of a direct view of right-handed CPL information in our revised Supporting Information. By covering a CLC reflective film with good spectral overlapping between the reflection band and emission peak, the chirality of LCPCs can be strongly amplified, and the g_{lum} values reach -1.58 for LCPC-S, -1.72 for LCPC-D, and -1.39 for LCPC-T. A visualized 3D information encryption is demonstrated in our revised Supporting Information. as Supplementary Figure 48.

“5) There are many sentence and word errors in the manuscript.”

Our response: We are sorry for these mistakes and the manuscript has been carefully checked and the errors have been fixed in our revised manuscript.

“6) Currently, the combination of fluorescent materials and chiral liquid crystals has been widely used to develop CPL-active materials for their further development of various functional applications (Nat. Photonics, 2022, 16, 174; J. Mater. Chem. C, 2021, 9, 12590; J. Phys. Chem. Lett. 2021, 12, 598; Nat. Commun. 2020, 11, 5659.). For instance, a g_{lum} value is up to 1.9 from perovskites in chiral liquid crystal-based soft helix devices for cryptology and anti-counterfeiting (Matter 2022, 5, 2319.).

Overall, the advantages of the CPL-activated photonic capsules prepared by the technical route of this work for information storage are not particularly prominent. There are many combinations to choose from.”

Our response: Thanks for this critical comment on our work. After carefully reading these references mentioned by the reviewer, we think our work has the following innovations compared with them,

-Phototunable CPL: The CPL materials reported by previous work exhibit either no responsive behavior, a temperature-dependent performance, or irreversible dual-color photoswitching. Our work demonstrates the excellent *phototunable CPL from blue emission (initial state) to full-color emission (photostationary state)*, even some of them show *reversible phototuning* (i.e. LCPC-D and LCPC-D/T).

-Facile processibility: Although CPL materials based on small-molecular CLCs have large g_{lum} values, their CPL strongly depends on the planar orientation of LC cells, and the liquidity also limits their applications. By encapsulating the CLC systems into LCPCs, this work *effectively addresses the issue that the small-molecular CLCs can be hardly processed. LCPCs can be easily processed on the flexible substrate by spin-coating or bar-coating, or act as the pen ink for direct writing.*

-Advanced applications: This work fully demonstrates the huge potential of LCPCs in the field of multi-level information encryption. Fluorescence, full-color, CPL, and time-dependence imparts LCPCs the capability of *four-dimensional encryption with a high-security level.*

As for the shortcoming that the reviewer considered as the low g_{lum} values, the g_{lum} values of CLC systems are on the order of 10^{-1} , of which the order of magnitude is consistent with those reported in the references. The extremely high g_{lum} value from perovskites in CLCs (*Matter* 2022, 5, 2319) is achieved by using the selective circularly polarized reflection of CLCs. Inspired by this, the amplified g_{lum} value of -1.72 for LCPC-D can be obtained by using a CLC reflective film with a green photonic band gap as mentioned before.

These mentioned references have been added into our revised manuscript to illustrate that CLCs have the latent capacity to generate high-performance CPL as Reference 23-25.

Reviewer 2:

“In the manuscript, the authors proposed a considerable 4D information encryption strategy based on liquid crystal photonic capsules (LCPCs), which realized the phototunable multicolor CPL through energy transfer and chiral transfer processes. Through mixing different component LCPCs, the full-color CPL films can be obtained, which can be successfully applied to multilevel information encryption. This study paves the way for advanced data storage encryption media and photonic devices. Anyway, I think that the present work is suitable for publication in Nature Communications after considering the following issues.”

Our response: We truly appreciate the respected reviewer for recognizing the novelty and significance of our research work, and kindly recommending this important work for publication in *Nature Communications*.

“1. The authors need to conduct NMR experiments to investigate the photocyclization/cycloreversion recovery process upon UV and visible light irradiation for DG and DR.”

Our response: Per suggestion, we have carried out NMR experiments to investigate the conversion rate of photocyclization/cycloreversion of DG and DR. We have added the description in our revised manuscript. The NMR spectra of DG and DR in different photostationary states together with related descriptions have been added in our revised Supporting Information as Supplementary Fig. 22-25.

“2. In this study, the CNB molecule was synthesized with (*s*)-1,1'-binaphthol and its right-handed photophysical properties were studied. And how about the (*R*)-1,1'-binaphthol, it is suggested that the authors discuss it.”

Our response: Thanks to the reviewer for this helpful suggestion. We have synthesized another chiral fluorescent switch with opposite handedness, designated as (*R*)-CNB, and “CNB” in our revised manuscript is all renamed as “(*S*)-CNB”. The properties of (*R*)-CNB in the LC host are systematically investigated, including (*R*)-LC-D and (*R*)-LC-T FRET systems. The corresponding description and discussion have been added into our revised manuscript. The synthesis and characterization of (*R*)-CNB have been added in our revised Supporting Information as Supplementary Fig. 13-17 and 18, 19, 26, 27, 29 and 30.

“3. In Fig. 2f, the blue line has a weak emission band near 600nm, the authors should retest the spectrum to eliminate misunderstanding.”

Our response: Per suggestion, we have tried to retest the CPL spectrum of LC-T system to eliminate the weak emission peak around 600 nm at initial state, but failed because DR is highly sensitive to the 365 nm excited light and isomerizes to *closed*-form during the test. To verify that LC-T system only exhibits blue emission at initial state, the CPL spectra are recorded by switching excited light with different wavelength (*i.e.* UV light and visible light that does not induce the photocyclization of DR). The CPL spectra of LC-D and LC-T systems have been added in our revised Supporting Information as Supplementary Fig. 31. The relevant discussion has also been added in our revised manuscript.

“4. On the page 7, for the sentence “of which the efficiency is increased by 12.7%”, the authors need to provide the data sources in detail.”

Our response: The fluorescence lifetimes of (*S*)-CNB are measured as 4.601 ns in the 5CB host, and 0.860 ns in (*S*)-LC-D system. According to the equation “ $E = 1 - \tau_{AD}/\tau_D$ ”, the FRET efficiency of (*S*)-LC-D is calculated as 81.3%. While the fluorescence lifetimes of (*S*)-CNB in (*S*)-LC-T system is measured as 0.278 ns, and the FRET efficiency of (*S*)-LC-T is calculated as 94.0%. When comparing the FRET efficiencies between (*S*)-LC-D and (*S*)-LC-T, the efficiency of (*S*)-CNB is increased by 12.7% after introducing the lower FRET step from DG to DR. All the supported data is shown in Supporting Information (Supplementary Table 1 and Table 2).

“5. Reversible photocyclization/cyclization under UV and visible light irradiation was confirmed in solution, and it is suggested to discuss the optical reversibility of LCPCs films through spectral experiments.”

Our response: Per suggestion, we have performed experiments on the phototunable reversibility in LCPC-D and LCPC-T films, and their fluorescence spectra as well as fluorescent images which have been added in our revised Supporting Information as Supplementary Fig. 37 and 38. The corresponding description has also been added in our revised manuscript.

“6. In Figure 4e, a series of colors are obtained by mixing different components of three color LCPC. It is suggested that the author display the corresponding component proportion of each state to make the results more scientific and rigorous.”

Our response: Per the suggestion, the blending ratios of different components of tricolor LCPC have been added to the fluorescent images in our revised manuscript as revised Figure 4.

“7. In Figure 13, “A combination of the letters” GDBQ “will be obtained”. There is no corresponding graph for “GDBQ”.”

Our response: The letter combination of “GDBQ” was placed to the right of the initial fluorescent image. To make it more striking, we have added the light-colored backgrounds to the decrypting message “GDBQ” and “FRET” in our revised manuscript as revised Figure 5.

“8. In Figure 5b, the concept of another dimension (polarization state) of information encryption is proposed, but it is not really realized. The author should emphasize the advantages of this work compared with previous reports.”

Our response: Per suggestion, we have simply summarized the recent progress of CPL materials in the field of information encryption in the introduction section in our revised manuscript. Using the CPL spectrum to identify the authenticity of the information has already been reported, but *multi-level information encryption using controllable CPL* is less developed. Compared with previous reports, the biggest advantages of our work are *4D information encryption* that first *introduces the time dimension into the CPL system* (color changing with time) and the *outstanding processability of LCPCs* that can easily coat on the flexible substrate or act as the pen ink for direct writing. Furthermore, much of the work focuses on the properties of CPL materials, while the concept of information encryption is demonstrated briefly. Our work reveals the huge potential of LCPCs applying for information encryption, which fully presents the encryption factors and decryption steps.

To promote the CPL from concept to practical application, a visualized 3D information encryption is designed using CLC reflective films to amplify the chirality of LCPCs which has been added in our revised manuscript as Supplementary Fig. 48. Taking the advantage of the selective chiral separation of the reflective films, the luminescent information is hidden under the left-polarized window, of which the emission peak overlaps well with the reflection band of the CLC film. Thus, the true information without the interference message can be directly observed using a left-polarized filter, rather than using analytical tools.

“9. In Figure 6, it is recommended to explain how the bar code corresponds to a digital code.”

Our response: Per suggestion, the instruction on how a bar code is converted into a digital code which has been added into our revised manuscript.

“10. The following paper, including research on multicolor luminescence and its application, can be referred to: *Angew. Chem. Int. Ed.* 2021, 60, 27171–27177; *Adv. Sci.* 2022,9, 2201523.”

Our response: Per suggestion, we have cited these two references in our revised manuscript as Reference 55 and 56.

Reviewer 3:

“In my opinion, the article is not suitable for publication in *Nature Communications* for the following reasons.”

Our response: We thank the reviewer for his/her time and comments on our work.

“The work is not innovative enough: There are many similar articles in the literature in which fluorescent molecules are used to perform data encryption, including also the ref.

50 by the same authors of this article; the only difference is that in ref. 50 thermally reconfigurable luminescent materials are used.”

Our response: Thanks to the respected reviewer for this critical opinion. However, we respectfully disagree with the expression “the only difference” between the thermo-responsive fluorescent molecules and phototunable fluorochromic materials, they show great differences in many aspects from molecular design, and material construction to even applications, where each difference demands creative thinking and innovative demonstrations. This work aims at building **CPL-active materials with phototunable full-color emission and mechanical stability**, not the common fluorescent materials. The color-tunability, polarization, and time-response endow such CPL materials the potential in the field of data encryption, because the phototunable CPL features multi-dimensional information and high-security level. The innovations of this work are as follows:

-Stepwise FRET: A photoswitchable CPL with **a large emission color migration of more than 160 nm** is achieved by introducing DG as an intermediate bridge within a FRET cascade from (S)-CNB to *closed*-DR.

-Phototunable full-color CPL: The **phototunable CPL from blue (initial state) to full-color (photostationary state)** is realized by constructing the RGB tricolor building blocks.

-Facile processibility: The difficulty of processing the small-molecular liquid crystals is solved by encapsulating the phototunable LC systems into polymeric shells. The obtained LCPCs can **easily be processed on flexible substrates by spin-coating or bar-coating methods, and can act as stable ink for direct writing.**

-4D information encryption: By creatively **introducing the time dimension into CPL materials**, LCPCs featuring **phototunable behaviors, full-color emission, and time-dependence** are constructed. These LCPCs exhibit huge potential in the field of 4D information encryption.

“Used physical systems are cholesteric capsules similar to those realized by other research groups that are completely ignored. Just to name a few of them:

1) Donato, M. G., Hernandez, J., Mazzulla, A., Provenzano, C., Saija, R., Sayed, R., ... & Cipparrone, G. (2014). Polarization-dependent optomechanics mediated by chiral microresonators. *Nature communications*, 5(1), 1-7.

2) Y. Li, J. J.-Y. Suen, E. Prince, E. M. Larin, A. Klinkova, H. Thérien-Aubin, S. Zhu, B. Yang, A. S. Helmy, O. D. Lavrentovich, and E. Kumacheva (2016). “Colloidal cholesteric liquid crystal in spherical confinement,” *Nat. Commun.* 7(1), 12520..

3) Petriashvili, G., Chanishvili, A., Zurabishvili, T., Chubinidze, K., Ponjavidze, N., De Santo, M. P., ... & Barberi, R. (2019). Temperature tunable omnidirectional lasing in liquid crystal blue phase microspheres. *OSA Continuum*, 2(11), 3337-3342, ISO 690
In particular, in the last one, the dependence of the cholesteric properties on temperature is used to realize a tunable laser.”

Our response: Thanks for the suggestion of the reviewer. In this work, LCPCs are prepared by the **interfacial polymerization** and have **polymer shells**, which are different from the self-assembled CLC microspheres in an aqueous solution. We have carefully read these excellent references for guidance on the preparation method of LCPCs. For example, PVA solution plays double roles during the process, one is an emulsifier to stabilize the emulsion, and the other is an orientation layer to induce the planar

arrangement of LC molecules. When demonstrating the arrangement of LC molecules in LCPCs, the effect of different diameters of CLC microspheres on the arrangement of liquid crystals is for consultation: the large diameter causes a concentric CLC layer packing; while the small diameter leads to the imperfect planar arrangement with little tangential orientation at the droplet periphery.

It is worth noting that the focus of this work is to *design a phototunable CPL material based on CLC*, and the photonic capsules are only a method to improve the processing performance. Thus, the characterizations of LCPCs are mainly focused on the CPL and fluorescent properties. According to these references, some characterizations of LCPCs have been supplied. For instance, the optical images of LCPCs under the crossed polarizers are observed to confirm that the liquid cores encapsulated in LCPCs are LC. The phase transition from CLC to the isotropic phase is observed on the hot stage at the heating speed of 1 °C/min. We have added the data and three references in our revised Supporting Information as Supplementary Fig. 35 and 36 and reference 63-65.

“Finally, the authors completely neglect aspects related to thermal, optical and mechanical stability of their systems. All these characteristics play a fundamental role in application oriented investigations of the proposed physical systems.”

Our response: Thanks for this helpful suggestion. The characterizations of thermal, optical, and mechanical stabilities of the photonic capsules have been carried out and have been added in our revised Supporting Information as Supplementary Fig. 36, 39. The corresponding description/discussion has been added into our revised manuscript. The information written by the LCPC inks exhibits excellent photo-, thermo- and mechanical stability, which can be clearly observed when being stored under ambient conditions for 2 months. In addition, it should be noted that the technical method of LCPCs in this study is similar to e-ink, both are microcapsules with polymer shells. Thus, LCPCs can also have good stability as discussed before.

Furthermore, we have carefully checked the revised manuscript and Supporting Information. With these changes and point-by-point response to the reviewers' comments as appropriate, we hope that the revised manuscript is now acceptable for publication.

Your kind consideration of this revised manuscript will be greatly appreciated.

Reviewers' Comments:

Reviewer #1:

Remarks to the Author:

I think the current work has been done very carefully and obtained results are interesting. Therefore, I recommend publication in Nature Communication.

Reviewer #2:

Remarks to the Author:

The authors have addressed my issues. Now I recommend its publication in Nature Communications.